# Unlocking the Potential of Therapy-Induced Cytokine Responses: Illuminating New Pathways in Cancer Precision Medicine

Dilip R. Gunturu [1,*], Mohammed Hassan [2], Deepa Bedi [1], Pran Datta [3], Upender Manne [4] and Temesgen Samuel [2]

1 Department of Pathobiology, College of Veterinary Medicine, Tuskegee University, Tuskegee, AL 36088, USA; dbedi@tuskegee.edu
2 Department of Biomedical Sciences, College of Veterinary Medicine, Tuskegee University, Tuskegee, AL 36088, USA; mhassan@tuskegee.edu (M.H.); tsamuel@tuskegee.edu (T.S.)
3 School of Medicine-Medicine-Hematology & Oncology, University of Alabama at Birmingham, Birmingham, AL 35233, USA; prandatta@uabmc.edu
4 Department of Pathology, University of Alabama at Birmingham, Birmingham, AL 35233, USA; upendermanne@uabmc.edu
* Correspondence: dgunturu@tuskegee.edu; Tel.: +1-334-727-8724

**Abstract:** Precision cancer medicine primarily aims to identify individual patient genomic variations and exploit vulnerabilities in cancer cells to select suitable patients for specific drugs. These genomic features are commonly determined by gene sequencing prior to therapy, to identify individuals who would be most responsive. This precision approach in cancer therapeutics remains a powerful tool that benefits a smaller pool of patients, sparing others from unnecessary treatments. A limitation of this approach is that proteins, not genes, are the ultimate effectors of biological functions, and therefore the targets of therapeutics. An additional dimension in precision medicine that considers an individual's cytokine response to cancer therapeutics is proposed. Cytokine responses to therapy are multifactorial and vary among individuals. Thus, precision is dictated by the nature and magnitude of cytokine responses in the tumor microenvironment exposed to therapy. This review highlights cytokine responses as modules for precision medicine in cancer therapy, including potential challenges. For solid tumors, both detectability of cytokines in tissue fluids and their being amenable to routine sensitive analyses could address the difficulty of specimen collection for diagnosis and monitoring. Therefore, in precision cancer medicine, cytokines offer rational targets that can be utilized to enhance the efficacy of cancer therapy.

**Keywords:** precision medicine; pharmacogenomics; cytokines; chemotherapy; radiotherapy





## 1. Precision Cancer Medicine

At the center of precision medicine, often interchangeably termed as 'personalized medicine' or 'individualized medicine', is the need to deliver 'the right drug, with the right dose at the right time to the right patient' [1,2]. Irrespective of the different interpretations of these terminologies, the overarching premise remains the recognition of individual patients' molecular differences as the basis to guide disease management, and consideration of individual patients as independent biological units to avoid 'group'-prescription of therapy and subsequent side-effects or non-responsiveness.

Currently, the most common application of precision cancer medicine involves pharmacogenomics, for which genomic determinants of drug metabolism (pharmacokinetics and pharmacodynamics) can be predicted from the polymorphisms and expressions of the enzymes that are involved [3–5]. Specifically, genomic information from normal or malignant cells is used to select an appropriate drug or patient, with the intent to avoid unnecessary treatment (Figure 1). Unlike proteomic, metabolomic, and other elements,

genomic data are the least variable, making it the most stable and widely comparable across multiple platforms.

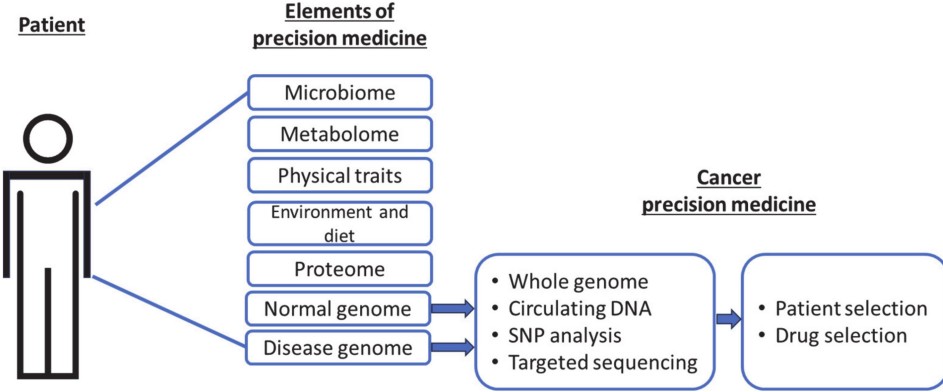

**Figure 1.** Current precision targets in cancer medicine as a subset of the broader elements of precision medicine.

It is practical and convenient to make clinical 'individualization' of patients based on the identification of genomic (DNA) or RNA signatures, and as such, to sort patients into 'most or least likely to respond', or 'most likely to experience side-effects' groups. However, precision medicine centered on DNA/RNA-based tools leaves a translational uncertainty, since proteins are the actionable targets and ultimate effectors of biological functions. This uncertainty becomes obvious when the proportions of patients 'expected' to respond are compared to patients who clinically 'respond.' Understandably, this uncertainty is multifactorial in origin and remains a challenge to identify and address the reasons for non-responsiveness, causing doubts about the entire precision approach [6,7].

## 2. Cytokine Pathways as Therapeutic Targets

Cytokines, cytokine receptors, and associated pathways offer opportunities to target signaling pathways associated with various disease conditions. Given their roles, inflammatory cytokines and receptors, as potential therapeutic targets, have been the focus of researchers, clinicians, and pharmaceutical companies (Figure 2). The most clinically advanced of these include antibody-based anti-psoriasis drugs targeting interleukin 23 (IL23, tildrakizumab), IL17A (secukinumab, ixekizumab), or IL17R (broadalumab). Additionally, anti-TNF antibodies have been deployed for the treatment of rheumatoid arthritis and psoriasis [8–10]. Among the most widely used drugs are an anti-TNFα antibody drug for arthritis marketed for humans as Humira, and an anti-IL-31 antibody for canine atopic or allergic dermatitis used under the trade name Cytopoint [11]. An IL6R-targeting mAb (tocilizumab) has been approved to treat a few inflammatory conditions, including rheumatoid arthritis and cytokine release syndrome after therapies [12]. However, the utility of tocilizumab against COVID-19 could not be established unequivocally, although IL6 levels correlated with severe disease [13,14]. IL6R antagonists, tocilizumab and sarilumab, are currently recommended for severely or critically ill COVID-19 patients (Therapeutics and COVID-19: living guideline. Geneva: World Health Organization [15]). Two cytokines, IL2 and IFNα, exhibited moderate therapeutic effects for the treatment of a variety of malignant conditions and were thus approved by the Food and Drug Administration (FDA). IL2 is used for advanced metastatic renal cell cancer and melanoma [16]; IFNα is indicated for hairy cell leukemia, follicular non-Hodgkins's lymphoma, melanoma, and AIDS-related Kaposi's sarcoma. In 2015, IL15 was approved for treatment of solid tumors [17].

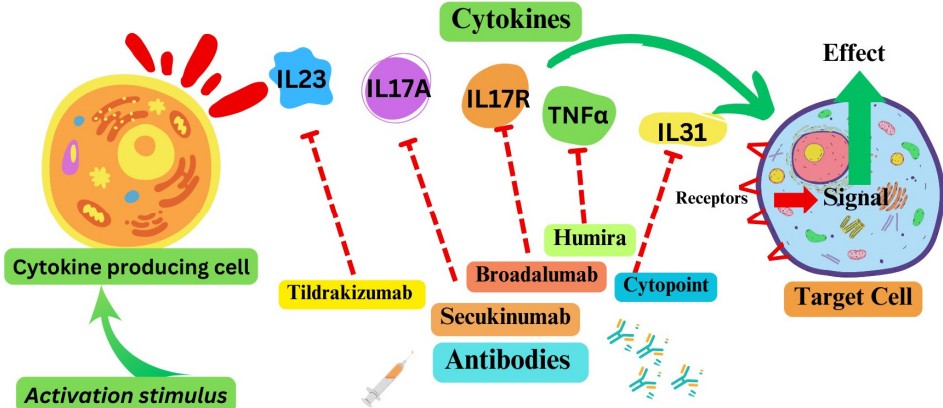

**Figure 2.** Some cytokine pathways as therapeutic targets. Note: The dotted red arrow represents the inhibition of the respective cytokine. Green arrows represent the activation signal.

Chemokines form a sub-group of cytokines characterized as molecules mediating the trafficking of immune cells [18]. Although several drugs targeting chemokine pathways have been developed, only a few have been approved for a disease condition. As an example, maraviroc, a CCR5 co-receptor antagonist that interferes with HIV entry into CD4 T cells, was approved in 2007 for treatment of HIV patients, in combination with other antiviral medications [19,20].

Currently, the primary goal in cytokine-directed therapy is to block the interaction of these molecules with the corresponding receptors, or to dampen the pathways downstream of the receptors. The impetus to advance the inhibitory approaches is because of the role of most of the targeted cytokines in enhancing disease processes, commonly inflammation. Unlike inhibition, an approach to enhance the expression or functions of cytokines for cancer therapy is still experimental, or is expected to be more challenging than inhibition, because of the need to fine-tune the activity of cytokines. For example, IL2, a potent T-cell mitogen, enhances antitumor responses of the immune system, but high doses can be detrimental to patients because of excessive cytokine release and activity [21].

## 3. Cells of Cytokine Origin

Most cell types within the tumor microenvironment, including tumor cells, immune cells such as macrophages, dendritic cells, and T cells, as well as stromal cells such as fibroblasts and endothelial cells produce cytokines in response to chemotherapy or radiation. Thus, tumor cells are involved in their own response to treatment. Macrophages, which are involved in the immune response, also produce cytokines in response to chemotherapy and radiation. Dendritic cells, another type of immune cells, also produce cytokines [22,23]; their role in cytokine production may involve the presentation of tumor antigens to T-cells, leading to an immune response. Antigen-stimulated CD4+ T-cells [24], along with natural killer T cells [25], CD8+ T-cells [26], mast cells [27], and dendritic cells [28] are the primary producers of IL2. Endogenous IL2 therapy increases the expression of CD25 and IL2 receptor, with subsequent proliferation of CD8+ T-cells [29]. IL2 enhances the expression of LAMP-1 [30] on the surface of CD8+ T-cells while decreasing the expression of PD-1 [31], an inhibitory receptor, thereby facilitating CD8+ T-cell cytotoxicity.

## 4. Cytokine Responses in Solid-Tumors Therapy: An Emerging Theme in Cancer Precision Medicine

As mentioned above, the concept of targeting cytokines or cytokine pathways to control specific diseases or processes is not new. After the role of receptors in viral entry was established, these pathways have even been targeted for infectious diseases such as HIV (CXCR4 and CCR5) [32–34]. Although most primary solid or intraepithelial tumors can be treated with several types of chemotherapy and radiation, adaptive resistance of

tumor cells and recurrence of cancer have hampered durable success of these therapies. The central theme of cancer precision medicine has been to identify patients who will favorably respond to a current therapy module, avoiding the side-effects of therapy on those who are least likely to respond [7,35].

For colorectal cancers, the clinical trials and applications of precision medicine have been narrowly focused on microsatellite status and the MAPK pathway proteins downstream of EGFR and HER2 receptors [36]; these are used for targeted therapeutics. Although the results and ultimate impact of these genomics-based approaches are still to be determined, to benefit patients, additional targets for precision medicine need to be identified. Microsatellite instability (MSI), defined by its genomic signatures and pro-mutagenic characteristics, is an appropriate tumor characteristic to identify colon cancer patients better suited for certain chemotherapy regimens, and especially for immunotherapy. However, a substantial proportion of the MSI tumors do not respond to current immunotherapy regimens [37]. Within the multifactorial background of 'non-responsiveness', which may involve added gene alterations, the role of signaling proteins may be substantial. After initiation of standard cancer therapy, a panel of targetable proteins unique to a patient could constitute a framework for precision medicine. Since the type and intensity of protein-response by malignant and normal cells in the tumor microenvironment will vary among individuals, the development and inclusion of 'therapy-induced cytokine signature' is emerging as a new dimension of precision medicine [38–40].

In malignant cells, cancer therapy using drugs or radiation elicits 'danger' signaling pathways, which have, primarily, a protective nature. These signals could be mediated by cytokines, and some cytokine pathways, such as transforming growth factor-beta (TGF-β), may have dual roles of inhibition or acceleration of tumor growth [41,42]. Other cytokine pathways may drive cell death, attract or deflect immune cells, or induce a senescence-like state to remain dormant through the stress period [43–46]. Therefore, an analysis of active cytokines in an individual's cancer microenvironment or tissues during chemotherapy, especially in comparison to the pre-intervention levels or activities, could reveal information about the biological mechanisms operating in the tumor under therapy stress. Such cytokine readouts may correlate with patient clinical outcomes [47–50]. Since cytokines are readily detectable in tissue fluids, this approach to precision medicine has an added value for clinical monitoring and decision making.

## 5. Cytokine Responses to Chemotherapy and Radiation

Cytokines are small proteins (peptides or polypeptides or glycoproteins with 5–30 kDa molecular weight) that are involved in modulation of the immune response. They are produced by various cells and act as signaling molecules, allowing for communication between different cell types. In the context of cancer, cytokines in the tumor microenvironment may be produced by both neoplastic and non-neoplastic cells that are involved in the regulation of tumor growth, proliferation, survival, and response to therapy. In recent years, there has been growing interest in understanding the role of cytokines in tumor response to chemotherapy and radiation [51,52].

Chemotherapy and radiotherapy induce various types of cytokines. These include pro-inflammatory cytokines, such as tumor necrosis factor-alpha (TNF-α), IL1, and IL6, as well as anti-inflammatory cytokines, such as IL10 and TGFβ [53]. Other cytokines, such as IFNs, chemokines, and growth factors, are also involved in tumor cytokine responses to chemotherapy and radiation. For example, IFNs enhance the antitumor immune response, and chemokines attract immune cells to the tumor site. Furthermore, growth factors, such as epidermal growth factor and vascular endothelial growth factor, are involved in angiogenesis, which is the formation of new blood vessels to support tumor growth [54]. Below, we provide examples of studies that assessed cytokine response after clinical or in vitro exposure of cells to chemotherapy or radiation (Table 1).

**Table 1.** Different studies on cytokine response after clinical or in vitro exposure of cells to chemotherapy or radiation.

| Study Reference | Cell Lines/Models | Treatment | Key Findings |
|---|---|---|---|
| [55] | MCF7 (BC), HCT116 (CRC) | Oxaliplatin, Cisplatin, 5-FU, Doxorubicin, Paclitaxel, Docetaxel, Carboplatin | Varying potential for prognostic significance in CRC vs. BC. Drug-specific and tissue-specific cytokine regulation after chemotherapy. Upregulation of TRAIL-R2 and chitinase 3-like 1 in BC cell line; downregulation of TRAIL and IFN-β. |
| [56] | CRC cell lines | Various chemotherapy drugs | Drug- and p53-dependent signatures, suggesting a personalized approach considering p53-status. |
| [57] | CRC cell lines | Topoisomerase inhibitor treatment | Induced cytokines clinically associated with patient overall survival. |
| [58] | MCF10A, MCF7, MDA-MB-231 | Ionizing radiation (9 Gy and 23 Gy) | Dose-independent, cell-line dependent release of cytokines and growth factors. |
| [59] | CRC models (in vitro and in vivo) | 5-FU chemotherapy | CCL20 induced after 5-FU, mediator of Treg recruitment and drug resistance. |
| [60] | Mice | Chemotherapy (Oxaliplatin) | Altered cytokine abundance in the hippocampus, potential toxic side-effects in the brain. Increased splenic populations of CD4, CD8, and Treg cells; altered splenic cytokine expression. |
| [61] | Not specified | Anthracyclines | Type-I interferon response, upregulation of CXCL10, resulting in antitumor outcome. |
| [62] | Solid cancers | Radiotherapy | Alters local or systemic cytokines. CXCL9, CXCL10, CXCL16 mediate anticancer effects; TGF-β, CCL2, CSF1, CXCL12, and insulin-like growth factor 1 may create an immunosuppressive microenvironment. |
| [63] | Solid cancers | Radiotherapy combined with cytokine treatments | CXCL9, CXCL10, CXCL16 mediate anticancer effects; other induced molecules may create an immunosuppressive microenvironment. |
| [64] | Rectal cancer patients | Chemoradiation therapy (CRT) | sCD40L and CCL5 levels associated with malignant tumor behaviors; higher post-CRT IL6 associated with a poor response. |
| [65] | Patients with bone and soft tissue tumors | Chemotherapy | Increases in IL6 and TNFα production, decreases in neutrophil counts after antitumor drug infusion. |
| [66] | Patients with hepatocellular carcinoma | Trans arterial chemoembolization (TACE) | IL-6 peaks on day 3, decreases thereafter; IL4, IL5, IL10 increase at two months after TACE. |
| [67] | Patients with NSCLC or GBM | CRT | TNF-α levels decrease in NSCLC, IFN-γ levels decrease in GBM after CRT. |
| [68] | Not specified | Gamma radiation-induced injury | Role of IL18 in response to gamma radiation-induced injury. |
| [69] | Mice, minipigs, nonhuman primates | Gamma radiation exposure | IL18 as a potential biomarker for radiation injury. |
| [70] | Tumor microenvironment | Radioimmunotherapy (177Lu) | Tumor regression not accompanied by a significant increase in attracting immune cell cytokines. |
| [71] | Not specified | Local high-dose radiotherapy | IFN-β production involved in ablative local radiotherapy-mediated tumor control. The antitumor effect of radiotherapy diminished in mice deficient in type I IFN response. |

A study comparing treated and untreated cell lines [55] evaluated the effects of clinical chemotherapeutics oxaliplatin, cisplatin, 5-fluorouracil (5-FU), doxorubicin, paclitaxel,

docetaxel, and carboplatin, using a panel of 52 cytokines in MCF7 breast cancer (BC) and HCT116 colorectal cancer (CRC) cells. The results showed that chemotherapy-inducible cytokine transcripts have varying potential for prognostic significance in CRC versus BC. For the BC cell line, the expression of TRAIL-R2 and chitinase 3-like 1 were upregulated by most of the drugs; TRAIL and IFN-β were downregulated. Beyond that, the two cell lines did not respond similarly, which suggested drug-specific and tissue-specific cytokine regulation after chemotherapy. Similarly, a comparison of chemotherapy drugs on cytokine production in CRC cell lines showed drug- and p53-dependent signatures, suggesting a personalized approach that considers the p53-status [56]. We also previously reported that topoisomerase inhibitor treatment of CRC cell lines induced cytokines that were clinically associated with patient overall survival [57].

In another study, the cytokine signature of conditioned media produced by the non-tumorigenic mammary epithelial cell line MCF10A, as well as by MCF7 and MDA-MB-231 BC cell lines, following single high doses of irradiation was examined [58]. The results showed that both 9 Gy and 23 Gy of ionizing radiation triggered the release of cytokines and growth factors capable of influencing tumor outcome in a dose-independent and cell-line-dependent pattern.

In an investigation that used both in vitro and in vivo CRC models, Wang et al. found CCL20 to be a major cytokine induced after 5-FU chemotherapy and a mediator of regulatory T cell (Treg) recruitment and drug resistance. Blockade of the cytokine by specific antibodies blocked tumor growth in vivo [59]. In another study with non-tumor bearing mice treated to measure the effects of 5-FU on cytokine expression in the brain, Groves TR et al. 2017 found that chemotherapy altered cytokine abundance in the hippocampus, with IL-1, -2, -3, -4, -5, -17, GMCSF (granulocyte macrophage colony stimulating factor), and RANTES expressed at higher levels in treated mice compared to controls, suggesting a mechanism for potential toxic side-effects of the drug in the brain. Oxaliplatin increased the splenic populations of CD4, CD8, and Treg cells, and altered splenic cytokine expression, despite decreasing the splenic mass [60]. Anthracyclines, used in chemotherapy for certain types of solid cancers, induced a type-I interferon response that upregulated CXCL10, resulting in an antitumor outcome [61].

Like chemotherapy, radiotherapy of solid cancers alters local or systemic cytokines [62], and radiotherapy may be combined with cytokine treatments for a beneficial outcome [63]. Nevertheless, unlike the antitumor mechanism of radiotherapy mediated by cell death, the contributions of cytokine-mediated functions of radiotherapy are not clearly understood. For example, although CXCL9, CXCL10, and CXCL16, in response to radiotherapy, mediate anticancer effects, other molecules, such as TGF-β, CCL2, CSF1, CXCL12, and insulin-like growth factor 1, which are also induced by radiotherapy, may create an immunosuppressive microenvironment [63]. After a sub-lethal dose of irradiation, enhanced levels of multifunctional cytokines IL1β, IL6, IL8, granulocyte colony-stimulating factor (G-CSF), granulocyte-macrophage-colony stimulating factor (GM-CSF), and TNF-α were recorded [48].

In a study aimed to characterize the predictive value of cytokines/chemokines for rectal cancer patients receiving chemoradiation therapy (CRT), platelets, immune system, and cancer cells, cross-linked through various cytokines/chemokines, were examined. The pre-CRT levels of soluble CD40-ligand (sCD40L) and the post-CRT levels of chemokine ligand-5 (CCL5) were associated with the malignant tumor behaviors of depth and invasion, and higher post-CRT IL6 was associated with a poor response [64].

In a separate study, serum cytokines (IL1β, IL2, IL6, and TNFα) were examined for a small number of patients undergoing chemotherapy for malignant bone and soft tissue tumors [65]. After intravenous infusion of antitumor drugs, increases of IL6 and TNFα production and decreases in neutrophil counts were evident. Similarly, when 13 cytokines were monitored after trans arterial chemoembolization (TACE) for patients with hepatocellular carcinoma, IL-6 reached a peak on day 3 before decreasing on and after day 7. In contrast, IL4, IL5, and IL10 increased at two months after TACE [66]. When the effect of CRT on

IFN-γ, IL6, and TNFα was investigated for patients with locally advanced non-small-cell lung cancer (NSCLC) or glioblastoma multiforme (GBM), TNF-α levels in NSCLC and IFN-γ levels in GBM decreased [67].

The role of IL18 in response to gamma-radiation-induced injury was reviewed [68]. The same authors also reported increases of IL1β, IL18, and IL33 expression in various organs, after exposure of mice, minipigs, and nonhuman primates to gamma radiation; they found IL18 as a potential biomarker to check radiation injury [69].

Elgstrom et al. [70] assessed the cytokine profile in the tumor microenvironment during tumor regression induced by radioimmunotherapy using an antibody conjugated to the beta emitter 177Lu. Although most cytokines in the tumor microenvironment were associated with immune cell infiltration, the differences in cytokine levels between treated and controlled tumors were small. The authors concluded that, for 177Lu radioimmunotherapy, tumor regression was not accompanied by a significant increase in cytokines that attract immune cells. In contrast, ablative local radiotherapy-mediated tumor control involved the production of IFN-β, and the antitumor effect of radiotherapy was diminished in mice deficient in type I IFN response [71]. The authors concluded that local high-dose radiotherapy initiated innate and adaptive immune attack cascades on the tumor through the type I IFN pathway.

## 6. Role of Cytokines Post Chemo- or Radiotherapy

In tumors, the cytokine response to chemotherapy or radiation serves several functions. First, pro-inflammatory cytokines such as TNF-α, IL1, and IL6 promote tumor cell death and inhibit tumor growth [72]. In tumor cells, these cytokines induce apoptosis, or programmed cell death, leading to a decrease in tumor mass. Second, pro-inflammatory cytokines attract immune cells to the tumor site and enhance their antitumor activity [73]. Third, cytokines modulate the immune response by promoting the differentiation and activation of immune cells [74]. Fourth, cytokines promote angiogenesis by stimulating the production of growth factors that induce the formation of new blood vessels in the tumor microenvironment [75]. In a recent review article, the authors reviewed the effects of different cytokines, such as IL-2, TGF-β, IL-4, IFN-γ, IL-6, and IL-23, on the tumor microenvironment and the CAR-T cell activity. They also described some strategies to overcome the immunosuppressive cytokines, such as dominant-negative receptors, inverted cytokine receptors, and T-cells redirected for universal cytokine killing (TRUCKs). They summarized the preclinical and clinical studies that have used different cytokines, such as IL-12, IL-15, IL-18, IL-21, IL-22, and IL-23, in combination with CAR-T cells [76]. In another recent review, the authors discussed the potential of cytokines as therapeutics for immune-related disorders. They highlighted the limitations of cytokines, such as their short half-lives and severe side-effects, and how innovations in bioengineering have aided in advancing our knowledge of cytokine biology and yielded new technologies for cytokine engineering. The authors presented an overview of therapeutics arising from cytokine re-engineering, targeting and delivery, mRNA therapeutics, and cell therapy. They also highlighted the application of these strategies to adjust the immunological imbalance in different immune-mediated disorders, including cancer, infection, and autoimmune diseases. Finally, the authors looked ahead to the hurdles that must be overcome before cytokine therapeutics can live up to their full potential [77]. Taken together, these cytokines can have pro-inflammatory effects, leading to tumor cell death, inhibition of tumor growth, and attraction of immune cells to the tumor site.

The reports briefly presented above point to the complex cytokine responses induced by chemotherapy and radiotherapy of diverse types of cancer. Elevated levels of specific cytokines may be associated with poor or favorable clinical outcomes; others may serve as potential theranostic biomarkers for treatment responses (Figure 3). Understanding the interplay between cytokines and receptors, cells of origin, and cancer therapy could lead to the development of new therapeutic approaches and personalized treatment strategies to improve patient outcomes. Further research in this field will uncover additional cytokine-related mechanisms and targets to enhance the efficacy of cancer treatment.

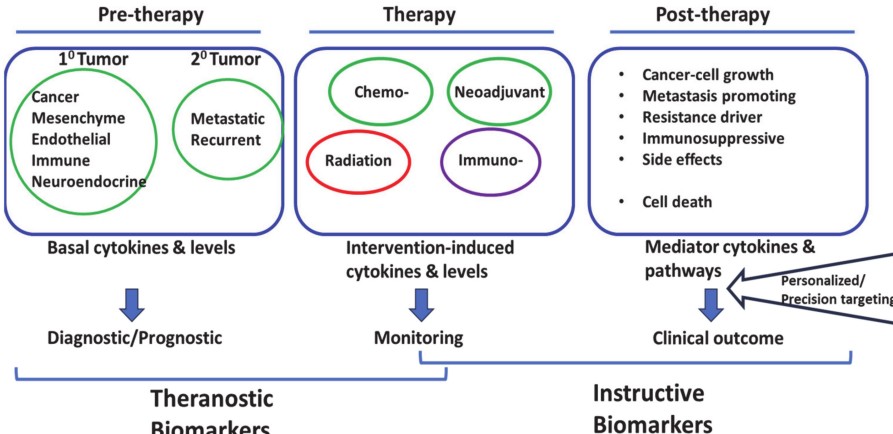

**Figure 3.** Post-therapy *instructive biomarkers* as potential targets for personalized precision approach.

## 7. Challenges in Cytokine-Directed Precision Approach

Unlike a pharmacogenomics approach to precision medicine, in which primarily genomic information is used to establish the choice of patients, cytokine response to any cancer drug may not be directly predictable from the patient genomic sequence. However, in a precision medicine approach, a specific cytokine response or pathway may be selectively or non-selectively triggered, enhanced, or suppressed, independently of the gene or transcript. Although cytokines or similarly functioning molecules induced by chemo- or radiotherapy may be considered for precision or personalized medicine, challenges inherent to cytokine biology and functions remain. Most cytokines are produced in small amounts, most act locally, and some of their functions may be transient [62]. Moreover, the ultimate biological outcomes of cytokine response may depend on the net additive or competitive functions of factors within the solid-tumor microenvironment space and time. Since the protocols and sensitivities of the methods to measure cytokines vary, across-the-board standardization or interpretation of results will pose challenges. Further, cytokines measured after therapy need to be calibrated against the downstream pathways activated by the trigger (therapy), to show their functional relevance as actionable biomarkers. Due to the overlaps and redundancies in cytokine-receptor interactions, as well as the conditional tumor promoting or suppressive dual roles of some cytokines, challenges will remain in predicting which outcome would dominate after therapeutic stress is induced in tumors. With these in mind, studies on cytokines as 'instructive biomarkers' to enhance cancer therapy need to be designed. Fortunately, advances in multi-omics and sensitive analytical methods could enable us to overcome such challenges.

Future research: With the analysis of therapy-associated cytokines introduced in this review, we propose future research can be directed to understand the interplay between cytokines, receptors, cells of cytokine origin, and cancer therapy. This will enable us to develop new therapeutic approaches and personalized treatment strategies to uncover additional cytokine-related mechanisms and actionable targets to enhance the efficacy of cancer treatment. We also propose exploring the use of multi-omics platforms and technologies to complement genomics-based cancer precision medicine to map cytokine profiles in tumor specimens or liquid biopsies, which can guide rational and targeted interventions.

Insights into the practical application and outcomes: The practical application of the concepts in this review includes clinical inclusion of cytokine responses as modules for precision medicine in cancer therapy, by identifying cytokines and their pathways as rational targets to enhance the efficacy of cancer therapeutic agents. Understanding the interplay between cytokines, receptors, cells of origin, and cancer therapy could lead to the development of new therapeutic approaches and personalized treatment strategies. We also draw attention to the clinical value of cytokine detection in tissue fluids for clinical monitoring and decision making after initiation of a standard therapy. Emerging and powerful technologies in research complement the existing genomics-based cancer precision

medicine with cytokine or related protein-based clinical decision strategies. Ultimately, actionable or instructive targets will be imperative to identify patients who could benefit from a secondary intervention after initiation of standard care therapy.

## 8. Conclusions

With the advent of multi-omics platforms and technologies, the use of genomics-based cancer precision medicine can now be complemented with cytokine- or related protein-based clinical decision strategies to find which patients could benefit from a secondary intervention after initiation of cancer therapy. This approach could open new avenues for the treatment of solid tumors for which options are limited, or for which low response rates or recurrences after chemotherapy or radiation hamper success. Maps of cytokine profiles in an individual's tumor specimen or liquid biopsy, with serial comparisons pre- and post-therapy, could guide rational and targeted intervention, which is the central tenet of precision medicine. This review discusses the application of precision cancer medicine, particularly in the context of pharmacogenomics, to identify patients who are most likely to respond to a specific therapy module and to avoid the side-effects of therapy on those who are least likely to respond. We also emphasize the need to understand the interplay between cytokines, receptors, cells of origin, and cancer therapy to develop new therapeutic approaches and personalized treatment strategies.

**Author Contributions:** Conceptualization, T.S., M.H. and D.R.G.; draft preparation, D.R.G., M.H. and T.S.; review, editing, D.R.G., T.S., D.B., P.D. and U.M. All authors have read and agreed to the published version of the manuscript.

**Funding:** Research in T.S., D.B., P.D. and U.M. labs is supported by NIH grant #U54CA118623.

**Acknowledgments:** The authors thank Donald Hill for editorial assistance.

**Conflicts of Interest:** The authors declare no conflict of interest.

## List of Abbreviations

BC—Breast Cancer; CCR—C-C chemokine receptor; CD—cluster of differentiation; COVID-19—Coronavirus Disease 2019; CRC—Colorectal Cancer; CRT—Chemoradiation Therapy; CXCL—C-X-C chemokine ligand; CXC—C-X-C chemokine; DNA—Deoxyribonucleic Acid; EGFR—Epidermal Growth Factor Receptor; FDA—Food and Drug Administration; GBM—Glioblastoma Multiforme; GM-CSF—Granulocyte Macrophage Colony Stimulating Factor; HIV—Human Immunodeficiency Virus; HER2—Human Epidermal Growth Factor Receptor 2; IFN—Interferon; IFN$\alpha$—Interferon-alpha; IFN-$\gamma$—Interferon-gamma; IL—Interleukin; mAb—Monoclonal Antibody; MSI—Microsatellite Instability; NSCLC—Non-Small-Cell Lung Cancer; RNA—Ribonucleic Acid; RANTES—Regulated on Activation, Normal T Cell Expressed and Secreted; TGF-$\beta$—Transforming Growth Factor-beta; TNF—Tumor Necrosis Factor; TNF-$\alpha$—Tumor Necrosis Factor-alpha.

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
