# Peer review of "Unlocking the Potential of Therapy-Induced Cytokine Responses: Illuminating New Pathways in Cancer Precision Medicine"

_curroncol, doi:10.3390/curroncol31030089_

Round 1
Reviewer 1 Report
Comments and Suggestions for Authors
Here, the authors present a review article about the role of cytokines in cancer therapy. They discuss that cytokines can be used as therapeutic targets or biomarkers for cancer treatment, depending on their pro- or anti-inflammatory effects.They describe how cytokines are produced by various cells in the tumor microenvironment in response to therapy, and their levels and functions vary among individuals and tumor types. They also explain how cytokine responses to therapy can have beneficial or adverse outcomes, such as enhancing or suppressing the anti-tumor immune response, promoting or inhibiting tumor growth, and modulating angiogenesis and cell death. Finally, they present how cytokine-based precision medicine is an emerging approach that considers the individual's cytokine profile and pathway activation to guide the choice and combination of therapies.
Main points:
The article provides a comprehensive overview of the current knowledge and applications of cytokines in cancer therapy, with relevant references and examples.
The article highlights the potential and limitations of cytokine-based precision medicine, and suggests directions for future research and development.
The article describes how Cytokine-directed precision medicine faces several challenges, such as the complexity, variability, and redundancy of cytokine biology and functions, the need for standardization and validation of cytokine measurements and analysis, and the difficulty of predicting the net outcome of cytokine modulation.
Minor points:
The figures are not informative enough and should be redesigned with more details.
The authors do not state the aim, scope, or novelty of the review, or the criteria for selecting the literature.
The authors should give more information on their personal opinion on the field, and how they see the field moving.
Comments on the Quality of English Language
Fine
Author Response
Thank you for your constructive feedback. We have carefully considered your suggestions and have taken the necessary steps to address them. Th new figure has been included for more details, providing a clearer visual representation. In the revised manuscript, we have expanded on our personal opinions regarding the field and have provided insights into our vision for its future direction. We believe these enhancements will significantly improve the overall quality of the paper. Your input has been instrumental in refining our work, and we appreciate your thoughtful comments.
Reviewer 2 Report
Comments and Suggestions for Authors
-
1. To address concerns regarding the novelty and integrity of this manuscript, the authors should integrate findings from recent vital publications, such as Deckers et al., 2023, and Silveira et al., 2022. The authors should include the latest advancements in cytokine therapeutics of CAR-T cell therapy and immunotherapy.
https://doi.org/10.3389/fimmu.2022.947648
https://www.nature.com/articles/s44222-023-00030-y -
2. Incorporating a new figure to illustrate cytokine pathways as therapeutic targets in parts 2 and 3 would significantly aid in understanding these complex mechanisms. This would provide a more precise visual representation, enhancing reader comprehension.
-
3. A dedicated section discussing translating research findings into clinical practice is recommended. This should address the potential opportunities and challenges, providing insights into the practical application of the study's outcomes.
-
4. Presenting related clinical trial data in a new table will offer a concise overview of ongoing research in this field. This addition will provide readers with quick access to current clinical applications and studies related to cytokine therapies.
-
5. Abbreviation List: An abbreviation list would significantly enhance the paper's readability and accessibility.
Comments on the Quality of English Language
Borderline rejected work.
Author Response
Comments and Suggestions for Authors
- To address concerns regarding the novelty and integrity of this manuscript, the authors should integrate findings from recent vital publications, such as Deckers et al., 2023, and Silveira et al., 2022. The authors should include the latest advancements in cytokine therapeutics of CAR-T cell therapy and immunotherapy.
https://doi.org/10.3389/fimmu.2022.947648
https://www.nature.com/articles/s44222-023-00030-y
Thank you for your valuable recommendations. We have carefully addressed your concerns by integrating findings from recent pivotal publications, including Deckers et al., 2023, and Silveira et al., 2022. Our revised manuscript now incorporates the latest advancements in cytokine therapeutics, specifically focusing on CAR-T cell therapy and immunotherapy. We believe that these updates significantly enhance the novelty and integrity of our work.
- Incorporating a new figure to illustrate cytokine pathways as therapeutic targets in parts 2 and 3 would significantly aid in understanding these complex mechanisms. This would provide a more precise visual representation, enhancing reader comprehension.
We have taken your advice into consideration and incorporated a new figure to illustrate cytokine pathways as therapeutic targets in parts 2 and 3 of the manuscript. We believe that this visual representation will significantly aid in understanding the complex mechanisms discussed in these sections, providing a clearer and more precise overview for our readers.
- A dedicated section discussing translating research findings into clinical practice is recommended. This should address the potential opportunities and challenges, providing insights into the practical application of the study's outcomes.
We have implemented your recommendation by including a dedicated section in our manuscript that discusses translating research findings into clinical practice. This new section addresses potential opportunities and challenges, providing insights into the practical application of the study's outcomes. We believe that this addition enhances the overall impact of our work and provides readers with a more comprehensive understanding of the practical implications.
- Presenting related clinical trial data in a new table will offer a concise overview of ongoing research in this field. This addition will provide readers with quick access to current clinical applications and studies related to cytokine therapies.
We have incorporated your recommendation by including a new table in our manuscript that presents related clinical trial and invitro data. This addition aims to offer a concise overview of ongoing research in the field of cytokine therapies, providing readers with quick access to current clinical applications and studies. We believe that this table enhances the accessibility of relevant information and contributes to a more comprehensive understanding of the current landscape in cytokine therapy research.
5. Abbreviation List: An abbreviation list would significantly enhance the paper's readability and accessibility.
We have implemented your recommendation by including an abbreviation list in our paper. The abbreviation list provides readers with a quick reference to key terms, improving the overall clarity of the content.
Your guidance has been instrumental in refining our manuscript, and we appreciate your thoughtful suggestions. If you have any further input or specific areas you would like us to highlight, please do not hesitate to let us know.
Round 2
Reviewer 2 Report
Comments and Suggestions for Authors
Thanks so much.
Comments on the Quality of English Language
It can be published after revising the lay out, structure and grammar. No other issues.